# Brief communication: Recent changes in summer Greenland blocking captured by none of the CMIP5 models

Edward Hanna [1], Xavier Fettweis [2], and Richard J. Hall [1]

[1]School of Geography and Lincoln Centre for Water and Planetary Health, University of Lincoln, UK
[2]Laboratory of Climatology, Department of Geography, University of Liège, Liège, Belgium

**Correspondence:** Edward Hanna (EHanna@lincoln.ac.uk)

**Abstract.** Recent studies note a significant increase in high-pressure blocking over the Greenland region (Greenland Blocking Index, GBI) in summer since the 1990s. Such a general circulation change, indicated by a negative trend in the North Atlantic Oscillation (NAO) index, is generally highlighted as a major driver of recent surface melt records observed on the Greenland Ice Sheet (GrIS). Here we compare reanalysis-based GBI records with those from the Coupled Model Intercomparison Project

5 (CMIP5) suite of global climate models over 1950-2100. We find that the recent summer GBI increase lies well outside the range of modelled past reconstructions and future GBI projections (RCP4.5 and RCP8.5). The models consistently project a future decrease in GBI (linked to an increase in NAO), which highlights a likely key deficiency of current climate models if the recently-observed circulation changes continue to persist. Given well-established connections between atmospheric pressure over the Greenland region and air temperature and precipitation extremes downstream, e.g. over Northwest Europe, this brings

into question the accuracy of simulated North Atlantic jet stream changes and resulting climatological anomalies over densely populated regions of northern Europe as well as of future projections of GrIS mass balance produced using global and regional climate models.

## 1   Introduction

Previous work notes strongly increasing mid-tropospheric high pressure over the Greenland region in summer over the past 2-3

decades (Fettweis et al., 2013; Hanna et al., 2015, 2018; McLeod and Mote, 2016). It is unknown to what extent this increased Greenland blocking, as measured through the Greenland Blocking Index (Fang , 2004; Hanna et al., 2013), has been triggered by low-level regional warming promoted by surface feedbacks (e.g. increased snow- and ice-melt, and Arctic regional sea-ice losses) as opposed to atmospheric dynamical (jet stream) changes; some recent studies (e.g. Francis et al. (2015)) suggest a slower-moving, more meridional northern polar jet stream, which may encourage more frequent and intense blocking over

Greenland. However, both of these mechanisms are likely to have played a role and moreover may well feed back off each other (Hanna et al., 2018). Increased Greenland blocking is a major contributor to the recent surface melt acceleration over the Greenland Ice Sheet (GrIS) because it favours the advection of relatively warm subtropical airmasses (Fettweis et al., 2013; Hanna et al., 2014; Delhasse et al., 2018) and promotes sunnier and drier weather conditions that enhance the melt-albedo feedback (Hofer et al., 2017). Changes in Greenland blocking are also important for mid-latitude weather and climate because

they perturb the North Atlantic atmospheric polar jet stream, where increased (decreased) blocking diverts the jet southwards (northwards) (e.g. Hanna et al. (2018)). Further recent work (Overland et al., 2012, 2015; Hanna et al., 2016; Hanna et al. , 2017) highlights Greenland as a key region linking the Arctic Amplification of global warming (Overland et al., 2017) with mid-latitude extreme weather, although such links are intermittent, itinerant and state-dependent, competing with a multitude
of other climate forcings (Hall et al., 2015; Overland et al., 2016). Previous work using climate-model projections to simulate NAO changes under sustained global warming conditions to 2100 finds a general slight – although not necessarily significant - trend towards a more positive future summer NAO (Gillett et al. , 2013; Fettweis et al., 2013). This contrasts with the observed trend towards a significantly more negative summer NAO since around 1990 (Hanna et al., 2015). However, although there is a strong antiphase between NAO and GBI changes (Hanna et al., 2013), this statistical relationship is of course not perfect, and
no similar model results of GBI changes have so far been presented.

Key outstanding research questions are: (1) what part of the Greenland atmospheric circulation anomaly (increase in blocking high pressure since around 1990) can be explained by decadal natural variability?; (2) how well is this natural variability represented in global climate models (GCMs)?; and (3) how will Greenland blocking frequency change in future? There is currently no clear consensus in the literature on these questions. Here we make concrete progress mainly on the first of these
15 questions by analysing current GCM simulations of Greenland blocking to see whether they capture the recent observed GBI changes, as a measure of how realistic these models may be for projecting future Greenland and North Atlantic regional atmospheric circulation changes. We conclude that there is a major disparity in trends between models from CMIP5 and observations for the last 20–30 years, suggesting that the projected future Greenland blocking decrease is probably unreliable, and that some key processes regarding blocking may be missing from the CMIP5 GCMs.

## 2    Methods and datasets

We calculated two "observed" GBI series based on NCEP/NCAR v1 Reanalysis 500 hPa geopotential height data (Kalnay et al., 1996). The first, which we here call GB1, is a simple area-weighted mean over 60–80°N, 20–80°W and follows previous work (e.g. Hanna et al. (2016)). We define a second GBI series, GB2, by subtracting the area-weighted mean GPH500 over the 60–80°N whole hemispheric zonal band from the area-weighted mean GPH500 over the standard GBI region defined above. This
is to allow for projected strong future Arctic warming raising geopotential heights over Greenland, which might mean that a future increasing GB1 mainly reflects increased atmospheric temperatures (Belleflamme et al., 2013) rather than a relative regional enhancement in blocking, where the latter is more directly depicted using GB2. The results of these calculations are shown in Figure 1 and show good agreement of trends and variability in both GB1 and GB2 changes for the recent record. Therefore we use GB2 for the rest of our analysis.
We also calculate a related air temperature parameter, TA2, which is defined as the mean free atmosphere temperature for the standard GBI region minus that over the hemispheric zonal band of 60-80ºN. TA2 is calculated using monthly temperature data at the 850, 700 and 500 hPa pressure levels from the monthly outputs as follows:

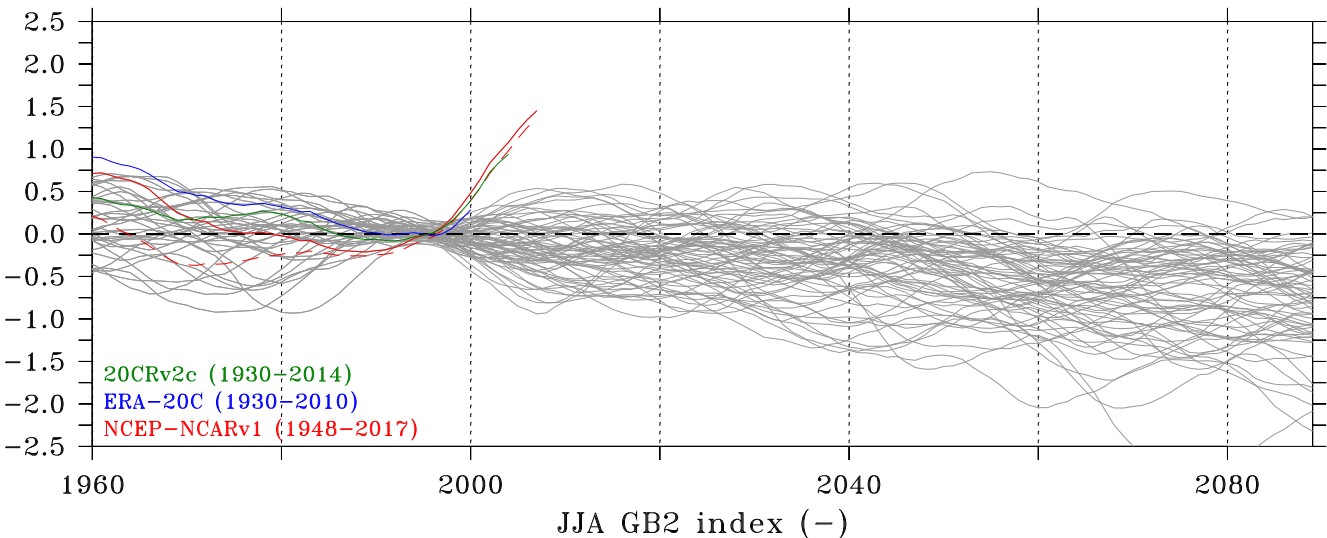

**Figure 1.** Time series of JJA GB1 (dashed red line) and GB2 (solid red line) indices over 1950-2100 as simulated by NCEP/NCAR v1 reanalysis (red line), by 20CRv2c reanalysis (green line), by ERA-20C reanalysis in blue as well as simulated by all the CMIP5 models (grey lines) for which both RCP4.5 and RCP8.5 scenarios are available. For the CMIP5-based time series, the Historical scenario is used over 1900-2005 and both RCP4.5 and RCP8.5 afterwards. A 20-year running mean has been applied to smooth the time series, and values have been normalised (average = 0 and standard deviation = 1) using 1986-2005 as reference period.

$$TA2 = (T850 + T700 + T500)/3 \qquad (1)$$

We use all Coupled Model Intercomparison Project 5 (CMIP5) GCM model outputs, for which both RCP4.5 and RCP8.5 scenarios are available, to simulate GB2 and TA2 changes over 1950-2100; retrospective model runs are used to simulate the 1950-2005 period, and all model output are based on standard (natural and anthropogenic) climate forcings. These data are

from CMIP5 run r1i1p1 of each GCM and therefore represent a single realisation of each one of the 36 GCMs from CMIP5 (see Table S1) and are not averages of ensemble members.All time series are smoothed using midpoint-centred 20-year running means (explaining why the first and last 10 years of time series are not shown) to emphasise long-term trends and variability linked to climate change, and to enable physically meaningful comparison of CMIP5 model output with the NCEP/NCAR v1 Reanalysis-based record which we use as a reference. Comparisons with the new centennial-timescale reanalysis, 20CRv2c

(Compo et al., 2011) and ERA-20C (Poli et al., 2016), are also added. However, as only surface data have been assimilated in these products, these reanalyses show biases in the free atmosphere, in particular in the summer free atmosphere temperature for which significant biases were found over Greenland by Fettweis et al. (2017) with respect to NCEP/NCAR v1 which is the only reanalysis shown here that assimilates soundings. This explains why the results of these long centennial reanalyses are slightly different from those obtained using the NCEP/NCAR v1 Reanalysis, even though of course all three reanalyses

represent exactly the same climate system. Finally, although 20CRv2c and ERA-20C cover the whole of the last century,

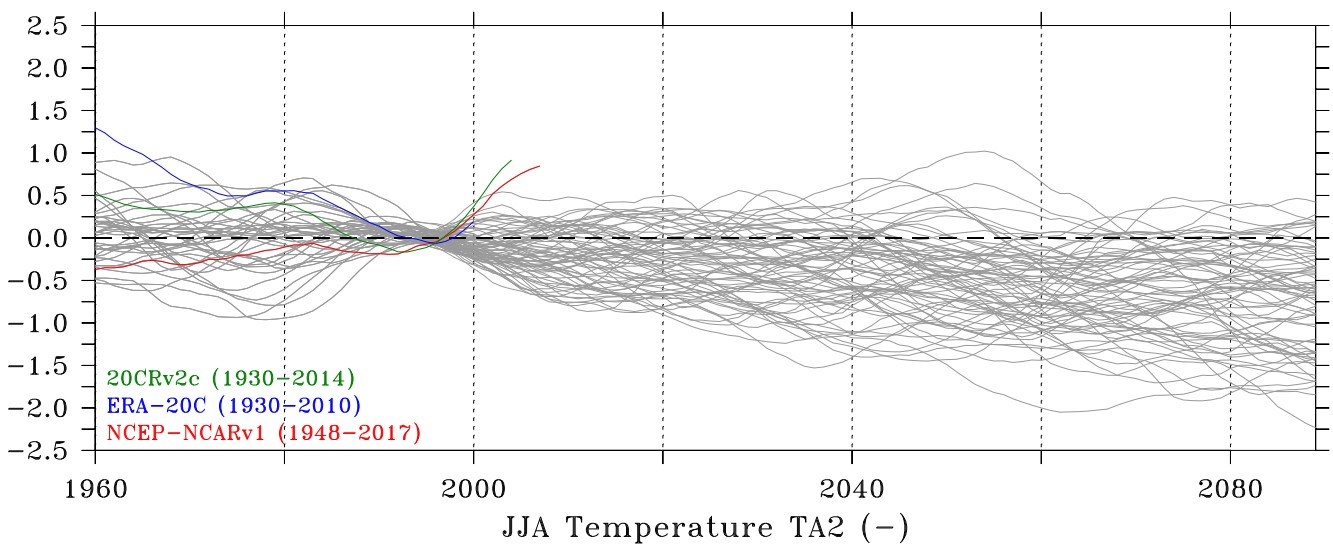

**Figure 2.** Similar to Figure 1 but showing TA2 (defined in Eq. 1). Values are normalised to the 1986-2005 reference period.

comparison with the observed record is limited here to 1950–2017 because this is the common period covered by the three reanalyses and because, as shown by Belleflamme et al. (2015), the general circulation of these centennial reanalyses diverges before 1940 over Greenland.

Finally, GB(X) and TA2 data are normalised using the recommended 1986–2005 recent past reference period (Hock, 2018).
All data and results used herein refer to the standard meteorological summer (JJA) season only.

## 3   Results

NCEP Reanalysis data since 1990, as well as both of the centennial reanalyses, show an increase in GB2 and normalised positive GB2 anomalies with a maximum reached at the beginning of the 2010's, which clearly exceed GB2 values projected by any GCM using both RCP4.5 and RCP8.5 as well as in the recent past GCM-based reconstructions using the Historical
scenario. Here, 20-year running means of GB2 time series are shown but the same conclusions can be drawn using either 30-yr or 3-yr running mean times series (see Figures S1 and S2 in supplementary material, while Figure S3 shows unsmoothed data). Based on unsmoothed annual data for 1996–2015, for example, the linear trend in NCEP GB2 is +1.70 m/yr and is statistically significant ($p<0.05$), while the mean linear trends in CMIP5 model runs are -0.03 (+0.02) m/yr with no individual model showing a positive trend greater than +0.97 (+0.96) m/yr for RCP 4.5 (8.5) respectively and with 64 of the 72 models
having trends within $\pm0.5$ m/yr (total CMIP5 sample size of $2 \times 36$). These results are confirmed for the slightly longer 1991-2017 period where we find a significant ($p<0.01$) trend of +1.32 m/yr for NCEP GB2 and mean (maximum positive) trends of -0.03 (+0.38) m/yr for CMIP5 RCP4.5 runs and +0.01 (+0.49) m/yr for CMIP5 RCP8.5 runs. Our results are insensitive to the choice of reference period (Figures S4 and S5). Likewise we find a recent marked increase in observed (reanalysis-based)

TA2 (see Figure 2) that is not replicated in any of the CMIP5 models: the latter show an overall reduction in GB2 and TA2, i.e. simulating fewer blocking events and weaker warming over Greenland compared with the rest of the Northern Hemisphere (Figure 2).

The disparity between the latest NCEP1 GB2 anomalies ($\sim +1.5$) and TA2 anomalies ($\sim +0.75$) ( (see Figure S6) indicates that the observed GB2 increase is unlikely to be fully driven by the Greenland regional free atmosphere temperature (TA2) increase, and there is also a recent (2010s) flattening off of TA2 while GB2 continues to increase. This leads us to invoke remote forcing from North Atlantic polar jet-stream changes advecting more southerly air masses over Greenland as being partly responsible. This effect is not shown in the CMIP5 model simulations, which project a near-uniform ratio of the normalised TA2 decreases to the normalised GB2 decreases. Neither is it shown in the centennial-timescale reanalyses but this is probably due to the absence of assimilation in the free atmosphere of these reanalyses and the associated biases in mid-troposphere heights and temperatures from 20CR and ERA-20C with respect to ERA-Interim and NCEP/NCAR v1 (Fettweis et al. , 2017). Finally, we note that while the NCEP/NCAR v1 based time series of TA2 ends with more stable (although still extreme) positive anomalies over the last couple of years, GB2 anomalies continue to increase over recent years and are not well simulated by any of the GCM-based time series.

## 4    Discussions and conclusions

Here we have shown that CMIP5 climate models do not adequately simulate the recent Greenland blocking increase since they project both a recent past and future decrease in Greenland blocking. We also note that the current observed positive blocking anomalies are significantly greater than simulated by any GCM for either current climate or future projections. Such models typically underestimate the magnitude of recent (since mid-1990s) Greenland warming, while previous work already suggested they are also not particularly effective at representing some key properties of North Atlantic jet-stream and blocking patterns (Davini and Cagnazzo, 2014; Davini et al., 2016; Hall et al., 2015).

The recent record rise in Greenland summer blocking may be influenced by the coincident positive phase of the Atlantic Multidecadal Oscillation (AMO), which is related to a more negative Summer NAO (Sutton et al., 2012; Folland et al., 2009) and therefore a more positive GBI. Since we are currently near the peak in the ( 80-year) AMO cycle, this effect could reverse in the next few decades, although – given the other drivers mentioned above – we consider this more likely to slow down the rate of GBI increase rather than result in decreased Greenland Blocking. Also, intrinsic atmospheric dynamics (internal variability) may have contributed to the recent GB2 increase, although there is likely to be a significant external forcing element too arising through Arctic/Greenland temperature feedbacks (Hanna et al., 2016, 2018). There is an issue of how well climate models capture internal variability in the GBI and NAO (Deser et al., 2017), and internal variability may result in different GBI trends in model output and observations for any given period of up to a few decades. However, the scale of the recent observed GBI change is well outside that represented in any of the CMIP5 models, and we do not subscribe to the view that most multi-decadal changes in these circulation patterns are mainly due to internal variability rather than being externally forced. Recent work reports limitations and negligible improvement in the last 20 years in model representation of Euro-Atlantic/Greenland

blocking that could be linked to limitations in available computer resource and/or to misrepresentation of the stratosphere and/or Atlantic sea-surface temperature patterns (Davini et al., 2016).

Our findings underscore the limitations of climate models in representing Greenland blocking, and so we question how realisticly the models represent North Atlantic circulation changes and hence European climatology: most notably winter temperature and windstorms and summer precipitation. Also the GCM-forced projections may underestimate future GrIS surface mass balance decreases by a factor of two, independently of the precise timing and amplitude of global warming, if the recent observed circulation changes continue to persist in summer (Delhasse et al., 2018). Model-observation discrepancies and thus model fidelity may, of course, be partly addressed in CMIP6 but clearly this is far from certain and meanwhile CMIP5 represents the current 'state of the science'. Given the recent rapid changes in Arctic climate and Greenland Ice Sheet dynamics – which were not well predicted 15-20 years ago – it is therefore essential that future climate modelling efforts focus on improving their representation of blocking, as this is a key aspect of mid-high latitude cryosphere-climate dynamics and change.

*Data availability.* Time series are available through a simple email request to the authors.

*Competing interests.* The authors declare that they have no conflict of interest.

*Acknowledgements.* X. Fettweis is a Research Associate from the Fonds de la Recherche Scientifique de Belgique (F.R.S.-FNRS). For their roles in producing, coordinating, and making available the CMIP5 model output, we acknowledge the climate modelling groups, the World Climate Research Programme's (WCRP) Working Group on Coupled Modelling (WGCM), and the Global Organization for Earth System Science Portals (GO-ESSP). We thank the European Centre for Medium-Range Weather Forecasts (ECMWF) for providing the ERA-20C (http://www.ecmwf.int) and the NOAA/OAR/ESRL PSD (Boulder, Colorado, US) for both NCEP-NCAR v1 and 20CRv2c Reanalyses (http://www.esrl.noaa.gov/psd/). Finally, we thank the two anonymous reviewers whose comments have significantly helped to improve the manuscript.

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
