# Peer review of "Brief communication: Recent changes in summer Greenland blocking captured by none of the CMIP5 models"

_The Cryosphere, 2018_

## Referee Comment (RC1) · Anonymous Referee #1 · 9 Jun 2018

Review of

Recent changes in summer Greenland blocking captured by none of the CMIP5 models

by Hanna and others

Major comments Recent studies, among which a significant number published by these authors, have demonstrated that the strong melt in Greenland post-1990 coincide with persistent summer blocking over the ice sheet (negative NAO index) resulting in the advection of warm air and relatively few clouds over the western ice sheet, both enhancing melt. Three all-important but quite distinct questions need to be addressed: 1) what part of that circulation anomaly can be explained by (decadal) natural variability; 2) how well is this natural variability represented in (global) climate models and 3) how

will the frequency of these the circulation patterns change in the future? Recent publications suggest that an important part of the circulation variability can be explained by natural variability with forcing originating in the tropics, but others claim that future warming will result in less zonal flow and hence more frequent blocking events, and that the recent anomalies are an expression of that. In order to add to this discussion, this paper should first carefully frame these issues, and indicate which question is addressed in this study. As the paper is organized now, it works confusing: is the main message that CMIP-5 models do not realistically represent Greenland blocking, and/or NH het stream dynamics in general? And does this imply that the projected future decrease in the blocking index is unreliable? And also that we cannot use these models to isolate the contemporary contribution of natural variability?

Another point of concern is the chosen averaging period to smooth the time series. Because the 20/30-yr running mean time series of the 'observed' quantities (Figs. 1/S1) still end close to 2020, we can conclude that the preceding decades are used to calculate the running means, i.e. it is no midpoint running mean. Were a midpoint approach used, the differences in these figures would likely be less dramatic. When looking at Fig. S2, in which three-year running averages are used, the difference between model and observations strikes me as less dramatic. Can the authors also show the annual (non-smoothed) JJA time series? From Fig. S2 it is evident that also for three-year running averages, the recent deviations exceed the intermodel variability for two periods, but recently the values have returned within the model envelope and a negative trend in the following decades appears still possible. Variability in the observed and model time series appear to be comparable. It would be instructive to calculate trend and standard deviation for all (unsmoothed) time series and see how they compare. And also: how sensitive are the plotted results to the choice of the reference period?

Minor (textual and technical) comments

p1, l. 6: please reformulate 'Historical scenario'.

p1, l. 15 and l. 17; p2., l. 16: please avoid citing the same documents multiple times.

---

## Author Comment (AC1) · 27 Jun 2018

We thank Referee 1 for their helpful comments. In the revised manuscript we will reformulate the introductory section to focus on the three questions, raised by the reviewer in his/her first paragraph, that need to be addressed. We are actually addressing all three of these questions. The main message is indeed that CMIP5 models do not realistically represent recent (last ~20-25 years) changes in Greenland blocking, and that this brings into question their reliability for future prediction of North Atlantic regional atmospheric circulation/dynamics changes.

Regarding the reviewer's concerns about data smoothing, we have re-calculated all the graphs based on mid-point running means and, as shown below, it hardly makes any difference to the results:

[Figure]

All data points are simply displaced a few years to the left but the relative changes and differences between modelled (CMIP5) and observed (reanalysis) GBI changes are the same. We will update this in the revised version. The CMIP5/reanalysis GBI trend difference is still clear even when using the annual (non-smoothed) JJA time series, as shown below:

[Figure]

We have also checked - and will show in the revised paper - that our results are insensitive to the choice of reference period. The following graphs are the same as Figure 1 but use 1960-1990 and 1960-2005 as the reference period. We also intend to present a comparison of trend and standard deviation statistics for all unsmoothed (CMIP5 and reanalysis) GBI time series in the revised version.

Edward Hanna and co-authors, 27 June 2018

Same as Figure 1 in paper but using 1960-1990 as the reference period:

[Figure]

Same as Figure 1 in paper but using 1960-2005 as the reference period:

---

## Referee Comment (RC2) · Anonymous Referee #2 · 13 Jul 2018

**Recent Changes in Summer Greenland blocking captured by none of the CMIP5 models**
Edward Hanna, Xavier Fettweis, Richard J. Hall

**Summary**
This paper presents timeseries of atmospheric blocking and temperature over the Greenland ice sheet simulated by global climate models from the Coupled Model Intercomparison Project 5 (CMIP5) for the period (1950-2100) and compares the modeled timeseries for the period 1950-2017 with the same fields from climate reanalysis data.  Atmospheric blocking and temperature patterns are quantified using a modified form of the Greenland Blocking Index (GBI), which is based on 500 hPa geopotential height anomalies over the area containing the Greenland ice sheet, and a temperature index derived for the same region.  The comparison indicates that none of the 36 global models examined are able to capture recent anomalous increases in GBI and temperature over Greenland relative to other areas in the same latitude band.  Nor do the models capture the magnitude of the recent positive anomalies.

**General Comments**
The paper documents an important difference between reanalysis datasets and global climate model outputs, highlighting that the climate models do not capture recent changes over Greenland that are believed to play a strong role in recent accelerating mass loss.  The paper is therefore highly relevant and of interest to the cryospheric and larger scientific community, and is well written.   The main question for the authors concerns whether the reanalysis data can really be established to be outside of internal model variability:

- Are the CMIP5 results averages of a set of ensemble members, or one realization from an ensemble?  If they are averages, wouldn't ensemble members show more variability, placing the reanalysis results within the internal model variability, making them consistent with a possible realization of some of the models?  If available, ensemble members should be included to evaluate this possibility.

**Specific Comments**
1. P. 2, Lines 10-11:  Suggest changing "climate model" to "global climate model" for clarity.
2. P. 2, Lines 27-31:  Please provide some more detail about the CMIP5 outputs.  Are these ensemble members or an average of ensemble members?  Briefly, what is the forcing applied to the historical simulations and future projections shown in Figs. 1 and 2.
3. P. 2, Line 29:  Please explain how the 20-year running mean was applied at the start and end of the timeseries
4. P. 4, Lines 7-14:  The disparities between TA2 and GB2 anomalies are much smaller for the other reanalysis products, especially for the period after 2000. This should be mentioned here.  Could this be because the NCEP reanalysis assimilates soundings as mentioned earlier, and therefore incorporate a process missing in the models?  Perhaps this could also help explain why the CMIP5 models do not appear to capture this relationship either.
5. P. 4, Line 11:  Would it be possible to show a plot of this ratio, perhaps in the supplementary material?

6. P. 4, Lines 17-18:  The future projections show larger negative anomalies.  Please clarify that positive anomalies are being referred to.  Also, it is hard to distinguish individual model lines, but it seems possible that comparable changes in the indices might be possible for a given 20-year period from some of the models (i.e. from a negative to a neutral state).
7. P. 5, Line 13:  Are there other reasons, besides the large magnitude of the change, for the view that these are not due to internal variability?
8. P. 5, Lines 4-24:  Can the authors suggest some possible mechanisms for what could cause errors in model representations of blocking?
9. Figure 1:  It is hard to tell where the GB1 index is plotted in Figure 1.  It would be helpful if an additional entry were added to the legend and the GB1 index were mentioned at the start of the caption instead of at the end.
10. Figure 2:  Note in the caption that values are normalized to the reference period.

**Technical Corrections**
1. P. 1, Line 19:  Change "(Francis et al. (2015)) suggests" to "(Francis et al., 2015) suggest"
2. P. 2, Line 2: Change (e.g. Hanna et al. (2018)) to (e.g. Hanna et al., 2018).
3. P. 4, Line 3:  Add reference to Figures S1 and S2 in the supplementary material.
4. P. 5, Line 16:  Perhaps change "how realistic is model representation" to "how realistically models represent"

---

## Author Comment (AC2) · 17 Jul 2018

We thank Referee 2 for their helpful comments. Regarding the referee's main concern, we can confirm that each one of the CMIP5-based Greenland Blocking (GB) time series is one realisation (run rlilpl) of each GCM from the CMIP5 database for which both RCP4.5 and RCP8.5 are available. Therefore it is fair to directly compare the variability of these series with that of the reanalysis GB series. In extra work we have now formally compared trends in the reanalysis and CMIP5-based GB series for several recent 20-year periods (1998-2017, 1997-2016, 1996-2015 and 1995-2014), which confirms that *all* of the CMIP5 GB trends are much smaller (about one order of magnitude less than) the reanalysis-based GB trends. Following our response to Referee 1, we will present a statistical analysis of these results, which will clearly demonstrate that the recently observed GB trend is well outside (larger) than any CMIP5 GB trend for the equivalent period. In the revised manuscript we will also address the specific comments (and make several technical corrections) raised by Referee 2.

Finally, this is a plot showing normalized TA2/GB2 as requested by Referee 2:

[Figure]

This indicates that since around the year 2000 TA2/GB2 is higher with NCEP1 than with other reanalyses where there is no assimilation in free atmosphere. But, as shown by Fettweis et al. (2017), there are biases in Z500 and T700 from 20CR and ERA-20C with respect to ERA-Interim and such biases explain the discrepancies between the reanalyses while NCEP1 and ERA-Interim agree well. Except for showing these discrepancies between reanalysis, we do not see the interest of such a plot.

A plot showing TA2 – GB2 is likely to be more useful:

[Figure]

This graph clearly confirms that the anomalies simulated by NCEP1 are out of range with respect to GCMs.

Edward Hanna and co-authors, 17 July 2018

---

## Author Response (AR1)

School of Geography
University of Lincoln, UK
22 July 2018

Dear Editor Prof. M. Tedesco,

We refer to the discussion on TCD for our responses to each individual reviewer.

In our revised manuscript the main changes, compared with the original version, are:

- clearer framing of the key outstanding research questions relating to Greenland blocking (see last paragraph of Introduction);
- clarification and more detail regarding the CMIP5 outputs (new text in paragraph 3 of Methods and datasets section);
- calculation and discussion of trend values for the last 20-30 years of all unsmoothed NCEP/NCAR reanalysis and CMIP5 Greenland blocking time series (in first paragraph of Results section);
- evaluation of the sensitivity of the plotted results to the choice of reference period (in first paragraph of Results section);
- consideration of possible mechanisms which may cause climate-model misrepresentation of blocking (end of second paragraph of Discussion/summary);
- the use of centred means in figures;
- extra figures in Supplementary Material showing annual (non-smoothed) JJA time series of Greenland blocking from reanalysis data and CMIP5 model runs, versions of Figure 1 based on different reference periods, and a time series of TA2 - GB2 (the latter in response to Referee 2).

All the minor corrections and improvements recommended by both reviewers have been made in the revised version of our manuscript.

Please don't hesitate to contact me if you need any clarification.

Best,
Edward Hanna and co-authors

[revised manuscript text omitted]